# National Antibiotics Utilization Trends for Human Use in Tanzania from 2010 to 2016 Inferred from Tanzania Medicines and Medical Devices Authority Importation Data

**DOI:** 10.3390/antibiotics10101249

**Published:** 2021-10-15

**Authors:** Raphael Zozimus Sangeda, Habibu Ally Saburi, Faustine Cassian Masatu, Beatrice Godwin Aiko, Erick Alexander Mboya, Sonia Mkumbwa, Adonis Bitegeko, Yonah Hebron Mwalwisi, Emmanuel Alphonse Nkiligi, Mhina Chambuso, Hiiti Baran Sillo, Adam M. Fimbo, Pius Gerald Horumpende

**Affiliations:** 1Department of Pharmaceutical Microbiology, Muhimbili University of Health and Allied Sciences, P.O. Box 65013, Dar es Salaam, Tanzania; hebysaburi52@gmail.com; 2Tanzania Medicines and Medical Devices Authority, P.O. Box 77150, Dar es Salaam, Tanzania; cassiamasa@gmail.com (F.C.M.); smkumbwa2003@yahoo.com (S.M.); adonisbitegeko@yahoo.com (A.B.); yonah.hebron@tmda.go.tz (Y.H.M.); emmaa_25551@yahoo.com (E.A.N.); adamfimbo@gmail.com (A.M.F.); 3Department of Pharmaceutics and Pharmacy Practice, Muhimbili University of Health and Allied Sciences, P.O. Box 65013, Dar es Salaam, Tanzania; bittycub@gmail.com; 4Department of Epidemiology and Biostatistics, School of Public Health and Social Sciences, Muhimbili University of Health and Allied Sciences, P.O. Box 65001, Dar es Salaam, Tanzania; alexandererick104@gmail.com; 5Department of Pharmacy, Kampala International University in Tanzania, P.O. Box 9790, Dar es Salaam, Tanzania; mhinachambuso53@gmail.com; 6Regulation and Prequalification Department, World Health Organization, 1211 Geneva, Switzerland; silloh@who.int; 7Department of Biochemistry and Molecular Biology, Kilimanjaro Christian Medical University College, P.O. Box 2232, Moshi, Tanzania; p.horumpende@kcri.ac.tz; 8Kilimanjaro Clinical Research Institute (KCRI), P.O. Box 2232, Moshi, Tanzania; 9Lugalo Infectious Diseases Hospital and Research Centre, General Military Hospital (GMH) and Military College of Medical Sciences (MCMS), P.O. Box 60126, Dar es Salaam, Tanzania

**Keywords:** antimicrobial, antimicrobial use, antimicrobial resistance, antibiotics utilization, Tanzania, Defined Daily Dose, Anatomical Therapeutic and Chemical Classification

## Abstract

Antimicrobial use (AMU) is one of the major drivers of emerging antimicrobial resistance (AMR). The surveillance of AMU, which is a pillar of AMR stewardship (AMS), helps devise strategies to mitigate AMR. This descriptive, longitudinal retrospective study quantified the trends in human antibiotics utilization between 2010 and 2016 using data on all antibiotics imported for systemic human use into Tanzania’s mainland. Regression and time series analyses were used to establish trends in antibiotics use. A total of 12,073 records for antibiotics were retrieved, totaling 154.51 Defined Daily Doses per 1000 inhabitants per day (DID), with a mean (±standard deviation) of 22.07 (±48.85) DID. The private sector contributed 93.76% of utilized antibiotics. The top-ranking antibiotics were amoxicillin, metronidazole, tetracycline, ciprofloxacin, and cefalexin. The DIDs and percentage contribution of these antibiotics were 53.78 (34.81%), 23.86 (15.44), 20.53 (13.29), 9.27 (6.0) and 6.94 (4.49), respectively. The time series model predicted a significant increase in utilization (*p*-value = 0.002). The model forecasted that by 2022, the total antibiotics consumed would be 89.6 DIDs, which is a 13-fold increase compared to 2010. Government intervention to curb inappropriate antibiotics utilization and mitigate the rising threat of antibiotic resistance should focus on implementing AMS programs in pharmacies and hospitals in Tanzania.

## 1. Introduction

Antimicrobial use (AMU) has continuously increased globally in the past decade. Though antimicrobial resistance (AMR) is a natural evolutionary phenomenon [1,2,3], AMU is one of the major drivers of the emergence of antibiotic-resistant microbes. This has been mainly caused by the dramatic increase in antibiotic consumption rates in low-and-middle-income countries (LMICs). The increase is partly due to the increased consumption of the “new and last resort” antibiotics, carbapenems, polymyxins, glycylcyclines, and oxazolidinones [4,5,6].

It has been well-established that AMU influences resistance [1]. Thus, changes in AMU patterns may be proxies reflecting changes in AMR patterns, which then influence the prescribing of antibiotics. In LMICs and higher-income countries (HICs), there has been a generally increased utilization of broad-spectrum antibiotics, including broad-spectrum penicillins, carbapenems, and polymyxins. However, the consumption of some of these classes of antibiotics such as cephalosporins, fluoroquinolones, macrolides, and second-line oxazolidinones have increased in the LMICs but decreased in the HICs [4,5]. The increase in the consumption of some antibiotics, such as third-generation cephalosporins, influences the emergence of extended-spectrum beta-lactamase-producing bacteria, which then confer resistance to other beta-lactam antibiotics [1,7].

Moreover, once present in the pool of bacteria, these resistant genes can hardly be removed and can be quickly passed to other bacteria through vertical or lateral transmission; they can even be passed to bacteria of different species [1,2,8].

The increase in AMU in LMICs is linked to rising economic growth as access to services and goods improves. However, economic growth is also ascribed to urbanization, facilitating infectious diseases such as enteric fevers, dengue, chikungunya, and viral diarrheal diseases, as well as increase in respiratory illnesses due to declining air quality [5,9,10,11]. All these influence antibiotics use, especially in the communities in LMICs where their access is poorly controlled. With a lack of stringent regulatory mechanisms, antibiotics are used with or without prescriptions for bacterial and viral infections. Furthermore, despite economic growth and access to antibiotics, this utilization is somewhat influenced by the social norms, cultural norms, and demands for prescribing, dispensing, and using antibiotics [4,12,13,14,15,16].

Tanzania is one of the fastest-growing economies on the African continent, with an average annual growth of 7% since 2000. Moreover, access to medicines has recently improved in Tanzania following the implementation of training to personnel authorized to dispense various medicines. This was designed to increase access to medicines and ensure that they are appropriately dispensed [17]. However, studies in Tanzania indicate a high burden of inappropriate use of antibiotics in communities that is mainly driven by a desire of medicine store-owners to make more profit and inadequate knowledge of the clients who pressure the dispensers. This clients’ pressure influences dispensers to abandon their ethics and practice inappropriate dispensing. Similarly, the inappropriate prescribing, dispensing, and using of antibiotics have been documented in hospital settings [10,12,18,19,20,21].

Consequently, resistance to the commonly available antibiotics has been rapidly and continuously increasing. Reports have shown the presence of up to 100% resistance of *Escherichia coli* and *Klebsiella pneumoniae* to commonly available penicillins and over 50% resistance to third-generation cephalosporins [22,23]. Almost 10% resistance to the “new and last resort” antibiotics such as meropenem has been documented in Tanzania. In addition, about two-thirds of staphylococcal isolates are methicillin-resistant *Staphylococcus aureus* (MRSA) [7,22,24].

The reports of inappropriate prescribing, dispensing and use of antibiotics originate from different research. There is a lack of national surveillance on AMU and AMR. Therefore, reports on overall antibiotic consumption can serve as a baseline evaluation of AMU for future efforts to control antibiotics use. This will further enable the trend analyses of antibiotics use and resistance over time, thus affirming the enforcement of policies to reduce antibiotics use in the country [5,25,26] and guide AMR stewardship (AMS) programs.

The standard method for estimating medicine utilization uses prescription data from hospitals or sales estimates from community medicine outlets. However, this method is not reliable in the Tanzanian setting, as it is limited by a lack or inaccuracy of records and poor data organization.

Medicine regulatory authorities, such as the Tanzania Medicines and Medical Devices Authority (TMDA), are responsible for regulating the importation of medicines. Thus, the accrued data can be used to estimate medicine utilization.

The need for accurate and more detailed antibiotic consumption data has led to the development of the Anatomical Therapeutic Chemical (ATC) and Defined Daily Dose (DDD) classification systems. These are used to measure medicine utilization based on the usual daily dose for a given drug, defined as the assumed average maintenance dose per day for a drug used for its main purpose in adults [27,28]. Therefore, in this study, we report the trend of antibiotic consumption in Tanzania based on human medicine importation data as a proxy for antibiotics utilization.

## 2. Results

A total of 14,301 records, 12,073 of which were of antibiotics for systemic use in humans, for antibiotics importation between 2010 to 2016 were reviewed. A total of 2228 records were excluded because they refer to antibiotics for either topical or veterinary use. A total of 154.51 DDD per 1000 inhabitants per day (DID) was utilized in Tanzania between 2010 and 2016 (Table 1), with a mean (standard deviation) of 22.07 (±48.85) DIDs.

The public and private sectors contributed 6.24% and 93.76%, respectively, of all DIDs of utilized antibiotics.

The oral and parenteral dosage forms contributed 151.18 (97.85%) and 3.33 (2.15%) DIDs (%), respectively (Figure 1). Upon further sub-dividing the antibiotics into respective dosage forms, capsules were found to comprise the major form of consumption of antibiotics (Appendix A).

When aggregating data according to ATC classification level 3, denoting chemical or pharmacological subgroups (Figure 2), we found that the top five most-utilized classes of antibiotics were penicillins (J01C); antibacterials (J01X); tetracyclines (J01A) and quinolones (J01M); and beta-lactam antibiotics (J01D), macrolide lincosamides and streptogramins (J01F). These top 5 classes contributed 97.55% of all consumption, of which 45.83% were contributed by the beta-lactam penicillins (J01C) class alone.

The beta-lactam antibiotics, penicillins (J01C), were the most utilized class of antibiotics according to level 3 ATC classification of the chemical or pharmacological subgroups. The individual antibiotics that made up the volumes of this class, in increasing order, were amoxicillin (J01CA04) (Figure 3A), ampicillin, ampicillin and cloxacillin, cloxacillin, phenoxymethyl penicillin, procaine benzylpenicillin, amoxicillin and clavulanate, benzathine penicillin, ampicillin and cloxacillin, benzyl penicillin, amoxicillin and flucloxacillin, flucloxacillin, sultamicillin, piperacillin and tazobactam, ampicillin combination, and ampicillin and sulbactam (Figure 3B and Appendix A).

Regarding the other antibacterials (J01X), metronidazole was the most highly utilized antibiotic in this class. Individual antibiotics that make up the volumes of this class, in increasing order of DIDs, are tinidazole, nitrofurantoin, ornidazole, linezolid, and vancomycin (Figure 4).

For the tetracyclines class (J01A), the order of increasing utilization was found to be tetracycline, doxycycline, oxytetracycline combinations, and chlortetracycline. For quinolones (J01M), the order of utilization was ciprofloxacin, levofloxacin, ofloxacin, norfloxacin, perfloxacin, sparfloxacin, nalidixic acid, and moxifloxacin (Appendix A). The order of other the beta-lactam antibiotics (J01D) class was cefalexin, cefuroxime, cefadroxil, cefixime, cefpodoxime, ceftriaxone, ceftazidime, cefaclor, cefotaxime, cefprozil, meropenem, cefepime, ceftriaxone combinations, cefoperazone combinations, and cefazolin (Appendix A). For macrolides, lincosamides, and streptogramins (J01F), the top antibiotics were erythromycin, azithromycin, clarithromycin, clindamycin, and roxithromycin (Appendix A). All the DIDs of individual antibiotics are ranked in Appendix A per year to show the annual trends.

According to the WHO AWaRe classification, the distribution of Access, Watch, and Reserve groups was 83.1%, 10.1%, and 0.008%, respectively (Table 2).

The combinations of antibiotics that are not recommended in the WHO AWaRe classification contributed up to 6.8%.

The model that included the data for the years 2010–2016 could significantly predict the future utilization of antibiotics (Figure 5). The ARIMA model could significantly (*p*-value = 0.002) predict the increase in utilization and forecast the trends of antibiotics up to 2022. The model estimated that by 2022, the total antibiotics consumed would reach 89.60 DIDs (Appendix A). This increase corresponds to about a 13-fold increase compared to the year 2010.

The top ten foreign antibiotics suppliers and local antibiotics importers with the highest consumed antibiotic DIDs are shown in Figure 6 and Figure 7, respectively.

The top supplier of antibiotics in the study period was from China, and eight of the top ten suppliers were from India and Kenya.

The Medical Stores Department was found to be the eighth top importer of antibiotics in the study period.

## 3. Discussion

The Global Action Plan of antimicrobial resistance stipulates the requirements of the surveillance of AMR and AMU to guide AMS in member countries [29]. In Tanzania, the implementation of AMS through the National Action Plan (NAP) on AMR started in April 2017 [26,30]. One of the critical elements of AMS is the monitoring of AMU in both animals and humans. We recently conducted a study to report the trend of AMU on antibiotics for veterinary use, and we showed that tetracycline, sulfonamides, trimethoprim, quinolones, aminoglycoside beta-lactams, and antibacterial combinations were the most frequently used antibiotics, with tetracycline on the top of the list in Tanzania [31]. The current study complements the trends of AMU in humans in a recent report by Mbwasi and colleagues [32]. The linear curve estimation for overall antibiotic consumption and the autoregressive integrated moving average (ARIMA) model that forecasted antibiotics utilization between 2010 and 2022 indicated an increasing trend of antibiotics utilization in Tanzania.

This is one of the few studies in sub-Saharan Africa that has attempted to estimate antibiotics utilization at the national level. Our data suggest an increase in the consumption of antibiotics, as reflected by a linear regression model. A total of 154.51 DIDs of antibiotics, with a mean of 57.4 (±48.85) DID (standard deviation), were utilized in Tanzania between 2010 and 2016. This average amount is slightly less than that reported for the years 2017–2019 in Tanzania [32], in which the mean consumption of antibiotics was 80.8 DIDs over three years. This difference could have been due to actual changing utilization patterns over the decades, while our study had only a shorter duration of observation. According to our data, there was an increase in utilization from 2010 to 2016. The utilization seemed to peak in 2017, according to a study by Mbwasi and colleagues [32], then declined in 2018 and 2019. The difference in the two reports may emanate from an actual decrease in the trends of the utilization of antibiotics or due to the additional data source used in the study by Mbwasi and colleagues [32].

A similar study performed in Kenya in 2004 showed a net decrease in antibiotic consumption [33]. Another similar study in Iran compared the utilization of antibiotics from 2000 to 2016 with Organization for Economic Co-operation and Development (OECD) countries in which antibiotic consumption ranged from 33.6 to 60 DID. The study noted a general increase in the consumption of antibiotics in certain classes of antibiotics such as sulfonamide and aminoglycosides [34]. A study comparing antibiotics utilization among European countries revealed an increase in antibiotic consumption by 36% from 2000 to 2010 [4]. In Sierra Leone, the total consumption of antimicrobials for the years 2017–2019 was 19 DIDs, which was much lower than the rates found in our study [35]. The current increase may correspond to the scaling of health insurance in the country, which has increased clients’ ability to access and purchase medicines [12,36].

Using an ARIMA model, we found a significant increase in antibiotics use in the study period in Tanzania. The model predicted that consumption of antimicrobials for 2017, 2018, 2019, 2020, 2021 and 2022 would be 55.09, 61.99, 68.90, 75.80, 82.70, and 89.60 DIDs, respectively. These predicted values are slightly more than those reported by the study of 2017–2019 [32] because the latter study noted a peak in 2017, followed by a decline in 2018 and 2019. Therefore, validating the ARIMA prediction model would require a dataset of more than seven years. The current model predicted that by 2022, antibiotics utilization would reach 89.90 DIDs, which is a 13-fold increase. This alarming excessive antibiotic consumption is likely to escalate AMR. Coincidentally, there are already many reports indicating the increase in AMR in Tanzania [22,24,37,38]. Therefore, these results call for a mechanism to control AMU. An urgent need to institute AMS in Tanzanian hospitals and pharmacies is warranted.

The current study shows that the public sector contributed 6.2% of all antibiotics in Tanzania between 2010 and 2016 (the Medical Stores Department (MSD), the government department for procuring medicines in Tanzania, mainly contributed to these public data). This was less than the 35% public sector contribution reported in a previous study [32] in Tanzania. The increasing contribution of the private sector in the sales of pharmaceuticals has been shown in other studies [32,39].

We found greater consumption of oral antibiotics (97.85%) compared to parenteral ones (2.15%) for systemic use. This may imply that there have been successful campaigns by the Ministry of Health on the safe use of oral products compared to injections, given that Tanzania has just begun implementing AMS [26,30]. This same reason may have applied to the decline of antibiotics utilization in 2018 and 2019 [32].

Upon aggregating the antibiotics utilization by ATC classification level 3 (chemical and pharmacological) subgroups, we found that the top five most-utilized classes of antibiotics were penicillins (J01C), antibacterials (J01X), tetracyclines (J01A), quinolones (J01M); beta-lactam antibiotics (J01D) and macrolide lincosamides and streptogramins (J01F). These classes contributed to 97.55% of all antibiotics utilization, with the beta-lactam antibiotics, penicillins (J01C), alone contributing 45.8% of all antibiotics utilized.

The five top-ranking individually utilized antibiotics in the study periods were amoxicillin (J01CA04), metronidazole (J01XD01), tetracycline (J01AA07), ciprofloxacin (J01MA02), and cefalexin (J01DB01). The DIDs and percentage contribution of these antibiotics were 53.78 (34.81%), 23.86 (15.44), 20.53 (13.29), 9.27 (6.0), and 6.94 (4.49), respectively. High and inappropriate use of amoxycillin has been reported in Tanzania [12,13,40], which may be the reason for increased utilization.

These top five antibiotics are recommended for various conditions in Tanzania Mainland’s Standard Treatment Guidelines (STG). Amoxicillin is recommended to treat acute respiratory infections [41], while metronidazole is used in anaerobic bacterial infections. On the other hand, tetracycline has been replaced by doxycycline in treating cholera, pelvic inflammatory diseases, and sexually transmitted diseases. Ciprofloxacin is mainly used for the treatment of urinary tract infections. Cefalexin is used in regional referral hospitals to replace penicillins [41]. The resistance levels to these antibiotics in this setting have been noted to be increasing [24]. The extensive use of amoxicillin in Tanzania may be attributed to its increased dispensing in accredited drug dispensing outlets (ADDO). The ADDO program provides training to dispensers, including the Integrated Management of Childhood Illness (IMCI), on, e.g., managing acute respiratory tract infections (ARIs) and diarrhea in children. The program provides guides for when antibiotics are needed for pneumonia and severe pneumonia only [18]. However, there are reports where antibiotic overuse and inappropriate antibiotics use have been observed [12,16,17,18] among ADDO dispensers. This may have driven the overuse of amoxicillin in Tanzania.

We cannot rule out that some medicines, such as tetracyclines may have been used in veterinary farming. Indeed, some samples obtained from animals and the environment in the Msimbazi River basin in Tanzania indicated high resistance levels to some antibiotics classes used by humans [40,42,43,44,45], such as tetracycline, nalidixic acid, ampicillin, and trimethoprim and sulfamethoxazole. These studies also revealed that farmers purchase these medicines without veterinary prescriptions [40,42,43,44,45].

In the class of beta-lactam antibiotics (penicillins), the other most consumed antibiotics were ampicillin and cloxacillin alone or in combination. In quinolone antibacterials, ciprofloxacin, levofloxacin, and ofloxacin contributed the most. In the class of other antibacterials, the most consumed agents were tinidazole and nitrofurantoin.

In the class of macrolides, lincosamides, streptogramins, erythromycin, azithromycin, clindamycin, clarithromycin, and roxithromycin were the only imported products.

There is a need to systematically investigate the prevalence of these highly consumed antibiotics in Tanzania since the data on AMR are still under investigation [1,7,22,24].

We also observed a higher dominance of suppliers of antibiotics from India. A similar trend has also been observed in other studies [39], indicating good trade in the sales of pharmaceuticals between Tanzania and India. However, the antibiotic quality determination was beyond the scope of this study.

This type of data can aid in the implementation of the NAP and steer national antibiotics AMS strategies [26].

### Limitation of the Study

We assumed that most medicines (80%) are generally imported by the Medical Stores Department (MSD). However, the contribution of public data from MSD was low in this dataset, which was retrieved exclusively from the TMDA. We also assumed that there are proper inventory control tools in medicine outlets to ensure that only a few medicines end up expired. Without these controls, clients could not utilize the entire amount of imported medicines presumed in the study.

Nevertheless, this study provides baseline data on antimicrobial drug usage, and it could help interpret any current or future emergence of antibiotic resistance. Further studies that account for local manufacturers and re-export records are warranted, as some imported medicines may be exported to other countries.

## 4. Materials and Methods

### 4.1. Study Design

This was a retrospective, longitudinal, analytical study that assessed consumption trends of standard units of antibiotics for human systemic use from the importation data on human medicine from 1 January 2010 to 31 December 2016 in the Tanzania mainland.

### 4.2. Study Setting

The United Republic of Tanzania, where this study was undertaken, is located at a latitude of 6.3690° S. and longitude 34.8888° E. Tanzania is bordered by eight countries and the Indian Ocean to the East. Uganda is found in the North, and Malawi and Mozambique are found to the South. To the southwest border is Zambia, and to the northeast border is Kenya. The Democratic Republic of Congo, Rwanda, and Burundi are located on the western border [31]. These serve as the port of entry of imported pharmaceuticals into Tanzania via seaport and airports. These ports include the Dar es Salaam airport and sea harbor (6.7924° S, 39.2083° E), Kilimanjaro airport (3.4245° S, 37.0651° E), Other ports are the national terrestrial border checkpoints in Sirari at 1.2512° S, 34.4763° E; Horohoro at 6.369° S, 34.8888° E; Namanga, Tunduma at 9.3096° S, 32.7689° E; and Mutukula at 1.0007° S, 31.4156° E [31].

### 4.3. Data Sources

The authors of this study used data on all antibiotics imported for human systemic use into Tanzania’s mainland obtained from the TMDA. The data included the port of entry identification at air, sea, and border checkpoints at Sirari, Horohoro, Namanga, Tunduma, and Mutukula, as well as the seaport in Dar es Salaam. Importation data can serve as a proxy to estimate the utilization of antibiotics because information on all medicines imported into the country is available. This method assumes that all medicines imported are utilized in Tanzania and that all medicines enter the country through the normal legal pathway, as regulated by TMDA. We also assumed that there is no illegal importation of medicines that goes undocumented [27]. Another assumption was that proper inventory control measures are generally taken to minimize expiry and losses in retail medicine stores. Almost all imported antibiotics are utilized for systemic use by humans.

The antibiotics used for systemic use, including those assigned the code J in the ATC system, were extracted. Within this group, there are several subgroups. The Defined Daily Dose is defined as the average dose required for maintenance.

Data were adjusted to the WHO/ATC classification system and expressed as several DDDs. DDD is a function of the total amount in grams of the antibiotics consumed, and the DDD for as a particular dosage form is shown by the following equation: [46]
((No of packages×No of tablets per package×No of g. per package))/((DDD of antimicrobial in grams)

The ATC classification system was used as our tool for medicine utilization research. This is a gold standard for medicine utilization research worldwide. The Defined Daily Doses of the antibiotics were calculated with the following formula.
(Total amount consumed in grams))/((DDD of antimicrobial in grams)

Reference was made to the DDD list available at the website of WHO Collaborating Centre for Drug Statistics Methodology [47] for the DDD values assigned to different antibiotics. Then, the amounts in Defined Daily Doses/1000 inhabitants-day were determined for each antibiotic, and the overall amount was determined using the function:DDD/1000 inhabitants-day = (Total DDDs in grams ∗ 1000)/(Population ∗ 365)

Reference was made to population estimates from the National Bureau of Statistics for the covered years (Appendix A).

### 4.4. Exclusion Criteria

All records, including topical products, such as lotion, cream, ointment, pessaries, shampoos and ophthalmic solutions, were excluded from the analysis. In addition, records lacking permit numbers, reference numbers, or permit issue dates were also excluded because their importation year could not be derived for the records to be included in the analysis. In addition, records including dates out of the study range were also omitted.

### 4.5. Data Collection

The TMDA is a National Medicines Regulatory Authority (NMRA) responsible for regulating the importation of medicines into the Tanzanian mainland market. The TMDA has outlined regulations and procedures that oblige importers to apply for an importation permit. After successfully evaluating the application, import permits are issued and archived using the TMDA’s Regulatory Information Management System (RIMS). Records were retrieved from the TMDA database of imported medicines.

The importation data collected include the date, generic and brand name of medicine, strength, category, quantity, the ATC classification of the antibiotic, company (suppliers or importer), price, currency, product manufacturer, and country of origin. Data were adjusted to the WHO ATC system and expressed as DDD measurement units. Utilization was expressed in DDD per 1000 inhabitants per day (DID). 

We also classified antibiotics according to the AWaRe classes. According to this classification, the Access group consists of antibiotics with activity against a wide range of commonly encountered susceptible bacteria with lower resistance potential than antibiotics in the other groups. The Watch class consists of antibiotics with higher resistance potential and includes most of the highest priority agents among the critically important antimicrobials for human use. The Reserve group includes antibiotics and antibiotic classes that should be reserved to treat confirmed or suspected infections due to multi-drug-resistant organisms. Reserve group antibiotics should be treated as “last resort” options [48].

### 4.6. Data Cleaning

Years were computed from dates, and where a date was missing, the reference number of the import permit that contains the year of approval was considered.

Data were reorganized to calculate milligrams (mg) and subsequent DDDs, as shown in the expression above. This involved rearranging pack size and strength to determine the total amount in mg for individual dosage units in different dosage forms.

Data were checked for accuracy, completeness, and reliability before analysis. The total grams of the utilized medicines were quantified by grouping the total amounts of active ingredients across the various formulations, thus accounting for the differences in strengths, dosage forms, and pack sizes.

We calculated the DDD of each antibiotic using a formula [49,50], in which the total amount consumed in grams was divided by the DDD of antimicrobial in grams. Reference was made to the DDD list available at the website of WHO Collaborating Centre for Drug Statistics Methodology [47]. The amounts in Defined Daily Doses/1000 inhabitants-day were determined for each antibiotic, and the overall amount was determined using the following equation:DDD/1000 inhabitants-day = (Total DDDs in grams ∗ 1000)/(population ∗ 365)

Reference was made to population estimates from the Tanzania Bureau of Statistics (TBS) to adjust the antivirals and antifungals consumption trends to the country population (Appendix A).

According to the AWaRe classes of antibiotics, we re-categorized the data proposed by the WHO [48,51]. The system comprises three groups of antibiotics: Access, Watch, and Reserve.

### 4.7. Data Analysis

Data files were combined, pivoted, and aggregated using Microsoft Excel 2013 (Microsoft Corporation, Redmond, Washington, DC, USA). The antbiotics’ strength, pack size, and quantity of the antibiotic were converted into milligrams, grams, and kilograms for further quantification of utilization. Generic names were harmonized to match the names in the ATC mapping file. We assigned each product a corresponding ATC code by matching it with the imported product’s generic name. This mapping allowed for the matching of the ATC and pharmaceutical categories. The amount in milligrams and DDD of an antimicrobial agent’s active ingredient was calculated and aggregated for each collected class. Tables and graphs were plotted to depict the trends in antibiotics utilization. Annual utilization data, aggregated per year, were entered into the Statistical Package for the Social Sciences (SPSS) version 20 (IBM Corp., Armonk, NY, USA). Time series and regression analyses were performed to ascertain the annual trend of antibiotics utilization. An ARIMA model was established to predict the trends of antibiotics use. A *p*-value of less than 0.05 was considered statistically significant.

## 5. Conclusions

Overall, our data indicate an increase in the actual and projected use of antibiotics. Beta-lactam antibiotics comprised 45.8% of all utilized antibiotics in Tanzania. Therefore, there is a need for a government intervention to curb inappropriate antibiotics utilization to mitigate the rising threat of antibiotic resistance. These interventions should be the focus of the National Action plan on AMR surveillance and antimicrobial stewardship programs.

In addition, a population-based prescription database for antibiotics may be developed for easy antibiotics prescription monitoring and information gathering for medicine utilization studies in Tanzania and other sub-Saharan African countries.

## Figures and Tables

**Figure 1 antibiotics-10-01249-f001:**
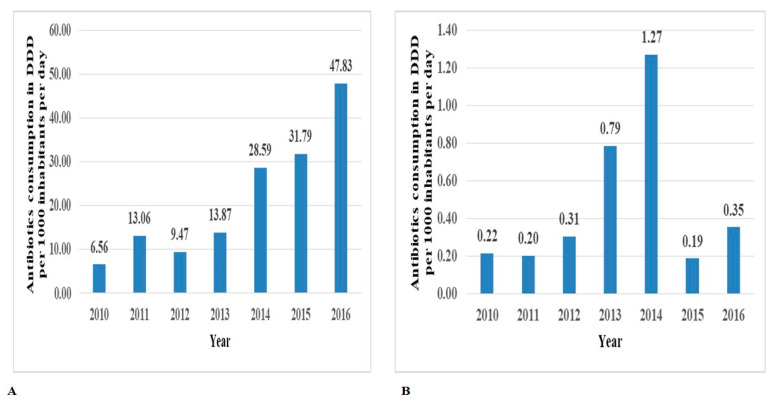
Contribution of oral (**A**) and parenteral (**B**) route of medicine administration for systemic antibiotics utilized over seven years.

**Figure 2 antibiotics-10-01249-f002:**
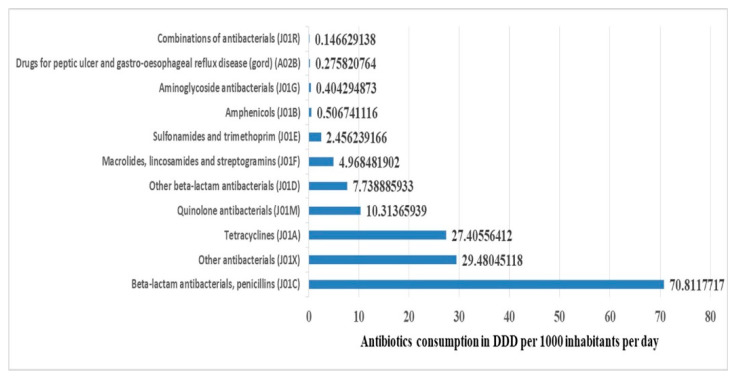
Contribution of each class (level 3 ATC classification) of antibiotics utilized in Tanzania from 2010 to 2016.

**Figure 3 antibiotics-10-01249-f003:**
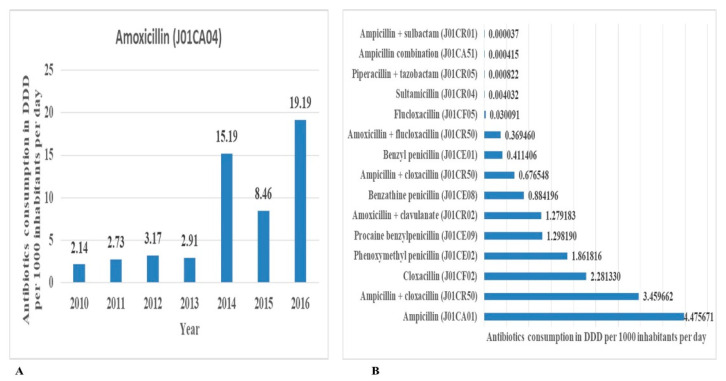
Contribution of each antibiotic in ATC class level J01C (**A**) for amoxycillin per year and (**B**) for the other antibiotics in class J01C utilized over seven years from 2010 to 2016 in Tanzania.

**Figure 4 antibiotics-10-01249-f004:**
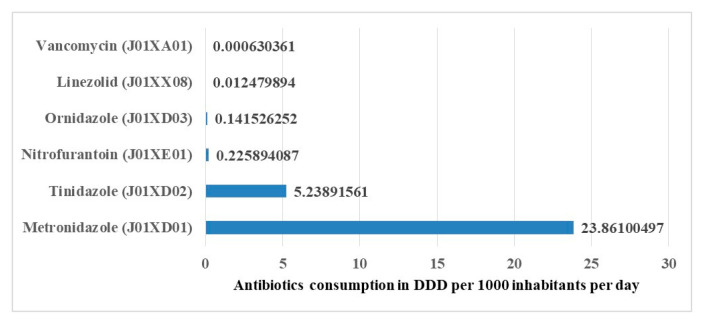
Contribution of each antibiotic in ATC class level J01X for the other antibiotics utilized over seven years from 2010 to 2016 in Tanzania.

**Figure 5 antibiotics-10-01249-f005:**
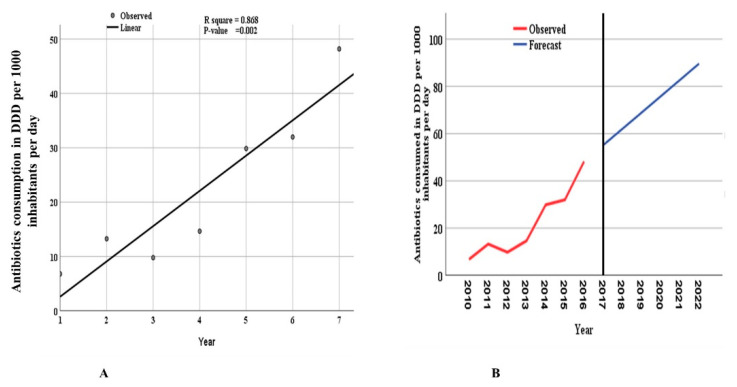
Trends of total consumed antibiotics over seven years from 2010 to 2016. Year 1 corresponds to 2010, and year 7 corresponds to 2016. The linear curve estimation for (**A**) overall consumption of antibiotics shows an increasing trend. (**B**) The autoregressive integrated moving average (ARIMA, 0, 1, 0) model forecasts utilization between 2010 and 2022.

**Figure 6 antibiotics-10-01249-f006:**
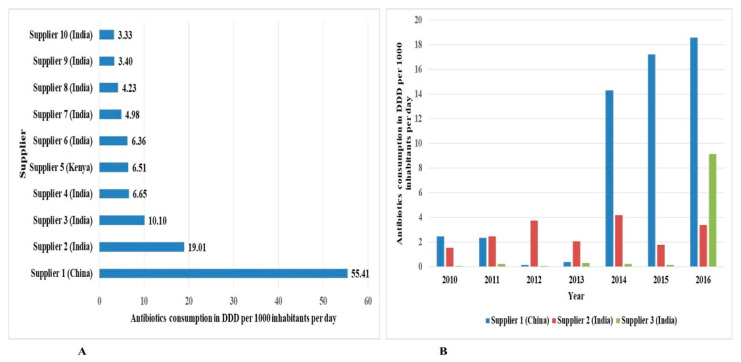
Top (local) importers of antibiotics utilized in Tanzania between 2010 and 2016. Panel (**A**) shows the top 10 importers, and panel (**B**) shows the annual distribution of the top 3 importers.

**Figure 7 antibiotics-10-01249-f007:**
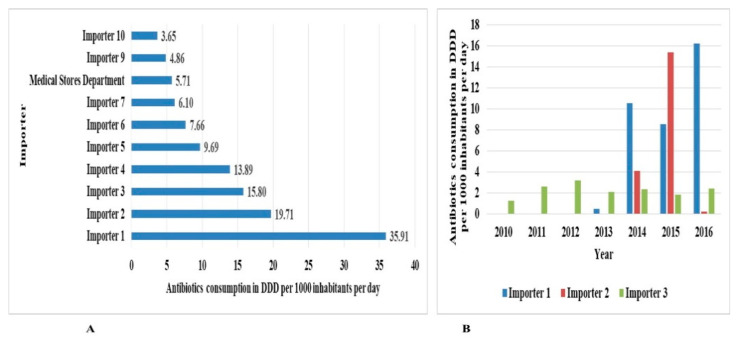
Top (foreign) suppliers of antibiotics utilized in Tanzania between 2010 and 2016. Panel (**A**) shows the top 10 importers, and panel (**B**) shows the annual distribution of the top 3 importers.

**Table 1 antibiotics-10-01249-t001:** Yearly distribution of DIDs and number of permits.

Year	Number of Permits	DID
2010	1154	6.78
2011	1578	13.26
2012	2008	9.78
2013	2000	14.65
2014	2136	29.86
2015	1551	31.98
2016	1646	48.19
Total	12,073	154.51

**Table 2 antibiotics-10-01249-t002:** Distribution of Defined Daily Dose (DDD per 1000 inhabitants per day (DID)) of antibiotics per the World Health organizations’ AWaRe class for antibiotics utilized in Tanzania from 2010 to 2016.

	Defined Daily Dose (DDD Per 1000 Inhabitants Per Day (DID)								
AWaRe Class	2010	2011	2012	2013	2014	2015	2016	All Year’s Total	% of Class
Access	5.060	11.238	6.307	10.049	25.382	28.879	41.456	128.371	83.083
Watch	1.253	1.454	2.445	2.000	2.655	2.020	3.729	15.554	10.067
Other	0.466	0.572	1.029	2.596	1.827	1.080	3.001	10.571	6.842
Reserve	0.0			0.010	0.002		0.001	0.012	0.008
Total	6.779	13.263	9.781	14.654	29.865	31.979	48.187	154.509	100.000

## Data Availability

The raw data supporting the conclusions of this article will be made available by the authors without undue reservation.

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
