# Peer review of "National Antibiotics Utilization Trends for Human Use in Tanzania from 2010 to 2016 Inferred from Tanzania Medicines and Medical Devices Authority Importation Data"

_antibiotics, 2021, doi:10.3390/antibiotics10101249_

Round 1

Reviewer 1 Report

The manuscript is interesting and well rganized, focused on the current topic of antimicrobial resistance,

The authors should better describe a proposal for a national antibiotic stewardship to make the manuscript ore interesting perspectively, also considering that they report a perspective view for the year 2022.

Use always italics for bacterial names (i.e., at lines 82-83)

Lines 154-155: The other antibacterials (J01X), metronidazole, was the most important utilized antibiotic other antibiotics. The phrase is unclear. What you mean?

Thorugh the text the acronym AWaRe is reported in different forms (AwaRe, etc.) , please revise and use the correct form only.

I suggest a complete revision of the text for English language and grammar.

Author Response

Reviewer 1 Comments

Dear Reviewer, Dear Editorial Office, below are responses to the comments raised about our manuscript. We have made changes as suggested by reviewers and highlighted yellow in the newly submitted document. We also responded to the comments as shown below.

Point 1: Moderate English changes required.

Response 1: We have edited the document to adjust the grammatical mistakes.

Point 2: Are the conclusions supported by the results (Can be improved)

Response 2: We thank the reviewer for this comment. We have now changed the conclusion to reflect our results.

Point 3: The manuscript is interesting and well organized, focused on the current topic of antimicrobial resistance,

Response 3: We are grateful to the reviewer for this positive remark and for taking the time to review our manuscripts thoroughly.

Point 4: The authors should better describe a proposal for a national antibiotic stewardship to make the manuscript ore interesting perspectively, also considering that they report a perspective view for the year 2022.

Response 4: We have argued about the use of the published data in the National Action Plan (NAP) for antimicrobial resistance (AMR) and we have also cited our reference on the implementation of AMR in Tanzania (Sangeda et al., 2020). The first and last paragraphs of the discussion have emphasized this fact.

Point 5: Lines 154-155: The other antibacterials (J01X), metronidazole, was the most important utilized antibiotic other antibiotics. The phrase is unclear. What you mean?

Response 5:  We have rephrased the sentence into "Regarding the other antibacterials (J01X), metronidazole was the highly utilized antibiotic in this class." We have also done the exact thing in subsequent similar sentences.

Point 6: Use always italics for bacterial names (i.e., at lines 82-83)

Response 6: we have re-written Escherichia coli and Klebsiella pneumoniae with italics.

Point 7: Thorugh the text the acronym AWaRe is reported in different forms (AwaRe, etc.) , please revise and use the correct form only.

Response 7: Thanks for the observation. We have changed the wrongly phrased occurrences of AWaRe as the correct form of the acronym.

Point 8: I suggest a complete revision of the text for English language and grammar.

Response 8: We have edited the document to adjust the grammatical mistakes.

Reviewer 2 Report

The study has been carefully designed and reported, limitations are acknowledged and the correct context regarding antimicrobial use is provided in the discussion. I have no major comments.

As a minor comment, I may suggest to provide readers with more details regarding the changes in resistance trends in Tanzania over the study period (if such data is available). Although the authors correctly reported prevalences of resistance to key antibiotics, it would be of interest also to know possible changes in resistance during the study years. 

Another minor comment regards the study period, which ended in 2016, making the study somewhat old. Would it be possible to retrieve further data from subsequent years to confirm the forecasted increase in 2022? (as we actually are now very close to 2022)

Author Response

Dear Reviewer, Dear Editorial Office, below are responses to the comments raised about our manuscript. We have made changes as suggested by reviewers and highlighted yellow in the newly submitted document. We also responded to the comments as shown below.

Point 1: English language and style are fine/minor spell check required.

Response 1: We have edited the document to adjust the grammatical mistakes.

Point 2: Are the conclusions supported by the results (Can be Improved)

Response 2: Response 2: We thank the reviewer for this comment. We have now changed the conclusion to reflect our results.

Point 3: The study has been carefully designed and reported, limitations are acknowledged and the correct context regarding antimicrobial use is provided in the discussion. I have no major comments.

Response 3: We are grateful to the reviewer for this positive remark and for taking the time to review our manuscript thoroughly.

Point 4: As a minor comment, I may suggest to provide readers with more details regarding the changes in resistance trends in Tanzania over the study period (if such data is available). Although the authors correctly reported prevalences of resistance to key antibiotics, it would be of interest also to know possible changes in resistance during the study years.

Response 4: We thank the reviewer for this comment. The data on the prevalence of resistance was not in the scope of this study. However, we have cited some references in the introduction showing the prevalence of different antibiotics resistance. A systematic study based on the currently highly consumed antibiotics is warranted. We have included such a statement in the discussion section that states, "There is a need to systematically investigate the prevalence to these highly consumed antibiotics in Tanzania since the data on AMR is still under investigation (Goossens, 2009; Mshana et al., 2013; Mikomangwa et al., 2020; Mnyambwa et al., 2021)"

Point 5: Another minor comment regards the study period, which ended in 2016, making the study somewhat old. Would it be possible to retrieve further data from subsequent years to confirm the forecasted increase in 2022? (as we actually are now very close to 2022)

Response 5: We have not managed to include the data for 2021. Since the data cleaning takes a bit of time, making the current data still old, the current data need to be published as a baseline for further studies. Nevertheless, we have compared our results with a study done in Tanzania using the same data source. However, for 2017-2019 (Mbwasi et al., 2020), some of our predictions indicate that to predict the future utilization, one needs several years' worth of training data, As short periods like 2017-2019 gave an impression of decreasing utilization. Whether this is correct remains to be tested with data from 2020 to 2022.

Reviewer 3 Report

General comments

The manuscript requires considerable language editing. Several sentences, especially in the results section do no make sense. Ensure that all scientific names are well written. For example, Lines 82 and 83.

In the discussion, there is more repetition of results than discussions on what the results imply. What accounts for the higher contribution of the private sector compared to the public sector? Also, the data showed and increase antibiotic use during the study period and the model predicts a further increase. What would account for such increase. It would be good to have some information regarding bacterial disease burden in Tanzania. And increase disease burden could lead to increased use.  However, the authors simply compare their study to their previous recent publication and indicate that the difference between the two studies could be due to increased utilization.

A major concern with the study is that the Tanzania Medicines and Medical Devices Authority records data on “all” pharmaceuticals imported into Tanzania. There is no indication in the write-up that a regulatory mechanism exists to ensure that the medications are for human or animal use. Also, without a regulation on prescription, it would be challenging to ascertain that all the drugs are not used for purposes other than human use. Tanzania is known for its growing veterinary industry and antibiotics are highly used by small and largescale aquaculture farmers for example. This points needs to be clarified.

Technically, the quality of all the figures needs to be improved.

Specific comments

 Line 31: DDIs or DIDs?

Line 63: The increased antibiotic consumption,…

Line 64-67: Here the authors link the increase antibiotic use to an increase in diseases that do not need antibiotics as treatment. This information is inaccurate. It would be good to mention bacterial infections and not parasitic and viral infections.

Line 72: 7% annually?

Line 78-79: “knowledge and training in rational dispensing”. Something is missing here.

Line 82-83: All scientific names in italics

Line 84-85: “Resistance….thirds of microbes”. This whole sentence needs to be rephrased.

Line 127-134:  This whole section needs complete rephrasing. The message passed here is not clear.

Line 136-137: “Figure 2: Contribution of each class (ATC level) antibiotics utilized over seven years (2010 to 2017).

Line 139-140: “The beta-lactam antibiotics, penicillins (J01C) was the most important level 3 ATC class that contributed the most”. This sentence needs to be rephrased.

Line 154-155: Rephrase

Line 343: Delete “both”

Author Response

Dear Reviewer, Dear Editorial Office, below are responses to the comments raised about our manuscript. We have made changes as suggested by reviewers and highlighted yellow in the newly submitted document. We also responded to the comments as shown below.

Point 1:  The manuscript requires considerable language editing. Several sentences, especially in the results section do no make sense. Ensure that all scientific names are well written. For example, Lines 82 and 83..

Response 1: We have edited the document to adjust the grammatical mistakes. In addition, we have re-written Escherichia coli and Klebsiella pneumoniae with italics.

.

Point 2: Are the conclusions supported by the results (Can be Improved)

 Response 2: We thank the reviewer for this comment. We have now changed the conclusion to reflect our results.

Point 3: In the discussion, there is more repetition of results than discussions on what the results imply. What accounts for the higher contribution of the private sector compared to the public sector? Also, the data showed and increase antibiotic use during the study period and the model predicts a further increase. What would account for such increase. It would be good to have some information regarding bacterial disease burden in Tanzania. And increase disease burden could lead to increased use. However, the authors simply compare their study to their previous recent publication and indicate that the difference between the two studies could be due to increased utilization.

Response 3: We thank the reviewer for this comment; in the discussion section

  • We have compared with other studies that show an increasing role of the private sector in the pharmaceutical supply chain in Tanzania.
  • We have indicated that use may be due to accredited dispensing outlets' approval to dispense antibiotics such as amoxycillin.
  • We have now included some references and reasoning for the increase, such as the scaling of national funds
  • We have included the references stating the levels of resistance to important antibiotics in Tanzania.

Point 4: A major concern with the study is that the Tanzania Medicines and Medical Devices Authority records data on "all" pharmaceuticals imported into Tanzania. There is no indication in the write-up that a regulatory mechanism exists to ensure that the medications are for human or animal use. Also, without a regulation on prescription, it would be challenging to ascertain that all the drugs are not used for purposes other than human use. Tanzania is known for its growing veterinary industry and antibiotics are highly used by small and largescale aquaculture farmers for example. This points needs to be clarified.

Response 4: Thank you to the reviewer for this concern.

We have explained in the methodology that the inclusion criteria were medicines intended for human systemic use. One of the exclusion criteria was antibiotics intended for veterinary use.

The TMDA regulatory mechanism specifies the category of use on import permits to indicate whether the medicines are for human or veterinary use,

However, how the medicines are used is beyond the control of TMDA. We can not rule out that some medicines such as tetracyclines may be used in veterinary farming. Indeed some samples obtained from animals and environments in the Msimbazi River basin in Tanzania indicated high resistance levels to some antibiotics classes used by humans, such as tetracycline, nalidixic acid, ampicillin, and trimethoprim and sulfamethoxazole. These studies also revealed that farmers purchase these medicines without veterinary prescriptions.

Point 5: Technically, the quality of all the figures needs to be improved.

Response 4: Thanks for the observation about the quality of the figures. We note that when the document is saved as PDF, there is a loss of image quality. We, therefore, re-attach the figures with the supplementary materials in jpeg format.

Point 5:  Specific comments

Line 31: DDIs or DIDs?

Response 5.1: We have changed all occurrences of DDIs to DIDs

Line 63: The increased antibiotic consumption,…

Response 5.2: We have changed the sentence into "The increase in AMU in the LMIC is linked to rising economic growth as access to services and goods improves"

Line 64-67: Here the authors link the increase antibiotic use to an increase in diseases that do not need antibiotics as treatment. This information is inaccurate. It would be good to mention bacterial infections and not parasitic and viral infections.

Response 5.3: We thank and agree with the reviewer that antibiotics are not meant to be used for non-bacterial infections. However, we have included the context where antibiotics are used for viral infections by adding a statement, "In lack of stringent regulatory mechanisms antibiotics are used with or without prescriptions for both bacterial and viral infections."

Line 72: 7% annually?

Response 5.4: We have rephrased the sentence to include annual … "Tanzania is one of the fastest-growing economies on the African continent, with an average annual growth of 7% since 2000."

Line 78-79: "knowledge and training in rational dispensing". Something is missing here.

Response 5.5: We have re-written the adjacent sentences to read, "However, studies in Tanzania indicate a high burden of inappropriate use of antibiotics in the communities, driven mainly by a desire of medicine store-owners to make more profit and inadequate knowledge of clients who pressure the dispensers. This clients' pressure influences dispensers to abandon their ethics and practice inappropriate dispensing. Similarly, inappropriate prescribing. dispensing and use of antibiotics have been documented in hospital settings (Morgan et al., 2011; Reynolds and McKee, 2011; Minzi and Manyilizu, 2013; Chilongola et al., 2015; Dillip et al., 2015; Mboya et al., 2018)."

Line 82-83: All scientific names in italics

Response 5.6: We have re-written Escherichia coli and Klebsiella pneumoniae with italics.

Line 84-85: "Resistance….thirds of microbes". This whole sentence needs to be rephrased.

Response 5.7: We have rephrased the sentences into "Almost 10% resistance to the "new and last resorts" antibiotics such as meropenem has been documented in Tanzania. In addition, about two-thirds of staphylococcal isolates were Methicilin Resistant Staphylococcus aureus (MRSA) (Mshana et al., 2013; Mikomangwa et al., 2020; Mnyambwa et al., 2021).

Line 127-134: This whole section needs complete rephrasing. The message passed here is not clear.

Response 5.8: we have rephrased the sentences into "According to ATC classification (Level 3 groups) (Figure 2), we found that utilization of the classes; beta-lactam antibiotics, penicillins (J01C), the other antibacterials (J01X), tetracyclines (J01A), quinolones (J01M), and other beta-lactam antibiotics (J01D) and macrolides lincosamides and streptogramins (J01F), constituted the top five of highly used individual antibiotics. These top 5 classes contributed 97.55% of all consumption, with the beta-lactam antibiotics, penicillins (J01C) alone contributing 45.83% of all antibiotics utilized."

Line 136-137: "Figure 2: Contribution of each class (ATC level) antibiotics utilized over seven years (2010 to 2017).

Response 5.10: We have rephrased the Figure 2 caption to "Contribution of each class (ATC classification level 3) of antibiotics utilized in Tanzania from 2010 to 2016."

Line 139-140: "The beta-lactam antibiotics, penicillins (J01C) was the most important level 3 ATC class that contributed the most". This sentence needs to be rephrased.

Response 5.11: We rephrased the sentence to "The beta-lactam antibiotics, penicillins (J01C), were the most utilized class of antibiotics according to level 3 of ATC classification".

Line 154-155: Rephrase

Response 5.12: We rephrased the sentences to remove ambiguity

Line 343: Delete "both"

Response 5.13: We deleted the word both

Round 2

Reviewer 1 Report

After the revision the manuscript is now acceptable for possible publication.

Reviewer 3 Report

Proofreading still required